# Ethical Dilemmas in Neonatal Care at the Limit of Viability

**DOI:** 10.3390/children10050784

**Published:** 2023-04-26

**Authors:** Lilijana Kornhauser Cerar, Miha Lucovnik

**Affiliations:** 1Department of Perinatology, Division of Obstetrics and Gynecology, University Medical Center Ljubljana, Zaloska 11, 1525 Ljubljana, Slovenia; 2Medical Faculty, University of Ljubljana, Vrazov Trg 2, 1000 Ljubljana, Slovenia

**Keywords:** extremely preterm infants, limit of viability, shared decision-making, end-of-life decisions

## Abstract

Advances in neonatal care have pushed the limit of viability to incrementally lower gestations over the last decades. However, surviving extremely premature neonates are prone to long-term neurodevelopmental handicaps. This makes ethics a crucial dimension of periviable birth management. At 22 weeks, survival ranges from 1 to 15%, and profound disabilities in survivors are common. Consequently, there is no beneficence-based obligation to offer any aggressive perinatal management. At 23 weeks, survival ranges from 8 to 54%, and survival without severe handicap ranges from 7 to 23%. If fetal indication for cesarean delivery appears, the procedure may be offered when neonatal resuscitation is planned. At a gestational age ≥24 weeks, up to 51% neonates are expected to survive the neonatal period. Survival without profound neurologic disability ranges from 12 to 38%. Beneficence-based obligation to intervene is reasonable at these gestations. Nevertheless, autonomy of parents should also be respected, and parental consent should be sought prior to any intervention. Optimal counselling of parents involves harmonized cooperation of obstetric and neonatal care providers. Every fetus/neonate and every pregnant woman are different and have the right to be considered individually when treatment decisions are being made.

## 1. Introduction

In past four decades, advances in medical technology have led to major improvements in neonatal care of premature children, and the limit of viability (at which some, but not all, neonates can survive) has shifted toward lower and lower gestations [1,2]. At the same time, however, the ability to prevent preterm birth has not changed significantly. As a result, the number of infants born at extremely low gestational ages has increased. Surviving very immature preterm infants have more early prematurity-associated complications and long-term neurodevelopmental deficits compared with more mature newborns [3,4,5,6,7,8]. This leads to clinical and ethical controversies on the role of obstetric antenatal interventions (e.g., steroid maturation therapy, use of tocolytics, antibiotic prophylaxis, magnesium sulphate neuroprotection, or site and mode of delivery), neonatal resuscitation, and continuation of life support [9,10,11,12,13,14].

Many clinicians question whether it is justified to treat this group of patients differently than other patients with similar prevalence of long-term disabilities (for example children with major congenital anomalies or severe infections or adults with certain critical illnesses). In a pilot study, Sanders et al. surveyed the attitude of US neonatologists in 1992 and reported that most of the respondents believed that non-intervention or compassionate care in the delivery room was appropriate for infants less than 24 weeks gestational age [15]. In Janvier et al.’s Canadian survey, only 35% of respondents would decide to treat a preterm neonate with a gestational age of 24 weeks, compared with 74% who would treat a 2-month-old child with meningitis, even though the survival and disability rates are equal in both patients [16]. These results led the authors to pose the question “why nobody likes preemies?” [16]. A similar survey of do-not-resuscitate decisions conducted in Slovenia and Croatia 10 years after the Canadian study showed that most students surveyed (as many as 97%) would decide to resuscitate even in the case of a preterm infant born after 24 weeks of gestation [17]. On the other hand, Rieder criticized the moral pressure to resuscitate by questioning the idea that resuscitation “saves” a life of a human in this context. In his opinion, resuscitation and treatment decisions in extremely preterm infants (born as early as 22 weeks) is controversial and the interventions are performed predominantly to take over “creating” it [18]. Similar reflections were shared by Vogelstein who proposed a framework for decisions regarding the best interests of extremely immature infants [19]. A recent survey of attitudes of Chinese obstetricians even reported on a very conservative approach towards the treatment of periviable infants as most obstetricians quoted that 28 weeks gestational age should be the cut-off value for providing full care to these infants [20].

Management of periviable neonates varies significantly [3,21,22,23,24,25,26,27,28]. A survey of 24 US tertiary perinatal centers showed substantial differences in clinical practices regarding the initiation of active treatment in infants born at 22 to 24 weeks of gestation [3]. The rate of active neonatal treatment at 22 weeks was 22.1% with a 95% confidence interval (CI) of 18.1 to 26.8. At 23 weeks, the rate of active treatment increased to 71.8% with a still wide 95% CI of 68.5–74.9. At 24 weeks, 97.1% of neonates received active treatment (95% CI 96.0–98.0). Importantly, differences in hospital rates of active treatment explained a large proportion of variability in survival and handicap-free survival of neonates born at the limit of viability among these centers [3]. The approach to management of preterm birth and neonatal resuscitation at 22 to 24 weeks of gestation varies substantially even among European countries [21,22,23,24,25,26,27,28]. Between-hospital and between-county variations in treatments of periviable neonates are in part caused by the fact that these aspects are rarely covered in national guidelines on perinatal practice [29]. Therefore, establishment of uniform ethical guidelines or recommendations for neonatal care at the limit of viability is necessary and would be very helpful for practicing obstetricians and neonatologists [30,31].

The aim of this paper is to provide a comprehensive review of the issues surrounding periviability and neonatal care.

## 2. What Is Periviability?

Periviability (also referred to as the limit of viability) is the earliest stage of fetal maturity (usually between 22 and 24 weeks of gestation) when there is a reasonable possibility, although not necessarily a high likelihood, of neonatal survival. Related terms include extremely low birth weight, extremely preterm birth, extreme prematurity, and borderline viability. The World Association of Perinatal Medicine (WAPM) defined a fetus to be viable when it is sufficiently mature to survive the neonatal period with available medical support. The prerequisite for the fetus to be considered a patient is that the pregnant woman receives obstetric antenatal care [32,33]. To accurately define the lower limit of gestational age for viability (e.g., 22–23 weeks), it would be vital to use consistent criteria. However, most of criteria used today are questionable. If we use mortality as such a criterion—which likelihood of survival is satisfactory? If we choose morbidity as a criterion—how severely must the child be impaired to be considered non-viable? Additionally, which injury counts more: that to the brain, the lungs, or one of the other organs [34]?

## 3. Ethical Concepts and Reflections

Ethics represents a fundamental dimension in the management of periviable births [35,36,37,38,39]. An ethical framework consists of obstetric and neonatal professional responsibility, which rejects rights-based reductionism (one-sided, narrow approaches to problems). It includes the ethical concept of the “fetus as a patient”. Medical personnel have obligations to the pregnant woman based on beneficence and respect of her autonomy as well as obligations to the fetus based on beneficence. This implies that the fetus is not an independent patient separate from the pregnant woman.

Beneficence (“doing good”) and the principle of nonmaleficence (“do no harm”) are among the oldest ethical principles in medical ethics (contained in the Hippocratic Oath), requiring the physician to find the best balance between clinical benefit and harm to the patient. Beneficence is the most important guide in efforts to save children who are incapable of life. On the other hand, the principle of “doing no harm” is critical when we decide to prolong a life associated with severe suffering.

Respect for autonomy (“respect for the individual’s will and choice”) is more complicated because fetuses/neonates do not have the autonomy of choice. Parents and clinicians have moral rights and legal obligations to make treatment decisions in the best interest of the child. It is crucial that physicians present pregnant women all the information about expected benefits and risks of potential medical interventions. Only by providing the pregnant woman (and often her partner) all the information can we empower her to make decisions and obtain her informed consent to continue or cease treatments.

The third ethical guideline is justice. This means not only that we treat similar cases of patients equally, but also that we use available resources effectively (equity of resource allocation). Aggressive treatment of an extremely immature preterm infant with a very low chance of intact survival can often be an inappropriate use of resources. Inequity, or even exploitation, occurs when a small proportion of patients receive clinical benefit while a much larger proportion does not simply because they did not have the opportunity to receive the same clinical treatment.

## 4. Clinical Guidelines

The World Association of Perinatal Medicine (WAPM) proposed an evidence-based approach to making decisions in obstetric and neonatal care for periviable fetuses/neonates. The Association’s recommendations are based on individual center data as well as on collaborative research. The latest guidelines are based on findings from the EPICure and MOSAIC groups, French EPIPAGE group, the United States National Institute of Child Health and Human Development (NICHD), and the Neonatal Research Network (NRN) [40,41,42,43,44,45,46]. Clinical guidelines for extreme preterm birth management based on similar data sources were also published by Guillén et al. in 2015 [47]. Approaches to antenatal/perinatal care where birth is imminent or indicated at less than 25 weeks of gestation is represented in Table 1.

There are robust data showing that extremely premature neonates have better outcomes when born in tertiary centers with neonatal intensive care units [47,48,49,50,51,52,53,54]. If possible, deliveries at the limit of viability should take place in centers that can provide the best professional care for the pregnant woman and the newborn, and that also have the appropriate infrastructure to support such treatments. Transfer of all pregnant women with threatened periviable preterm birth to a tertiary center must be encouraged (the so-called “transport in utero”). Delivery of a periviable neonate in a tertiary perinatal center with an adequately equipped and staffed neonatal intensive care unit can improve short-term and long-term neonatal outcomes [48,49,50,51,52,53,54]. In addition, an important proportion of periviable births are medically indicated due to severe maternal morbidity (infection, hypertensive disorders of pregnancy, e.g., HELLP syndrome, etc.). In these circumstances maternal transfer (“transport in utero”) will not only allow better neonatal care compared to postnatal neonatal transport but will also improve maternal outcomes and reduce risks of maternal mortality [48]. As a result, if at all possible, periviable births for which maternal or neonatal interventions are planned should occur in tertiary perinatal centers [48,53,54].

Tocolysis with atosiban, nifedipine, or indomethacin has been shown to be effective in delaying preterm birth for 48 to 72 h [55,56,57]. Although there is no evidence tocolytics allow longer pregnancy prolongation and fetal development, they can be considered in order to achieve maternal transfer to a tertiary center (“transport in utero”) and antenatal steroid administration [55,56,57]. It has to be noted, however, that most data on tocolysis effectiveness come from studies on preterm labors at more than completed 26 weeks of gestation [55,56,57]. There is very little evidence on their benefits in periviable gestations.

The recommendation on antenatal steroids in periviable births are based on laboratory studies on lung development and clinical observational studies [58,59,60]. Data from a 2011 Eunice Kennedy Shriver NICHD Neonatal Research Network study showed a reduction in death and neurodevelopmental impairment in children born at 23–25 weeks of gestation who were exposed to steroids in utero [58]. Importantly, no significant difference was reported in infants born at 22 weeks [58].

Broad spectrum antibiotics have been shown to prolong pregnancy and reduce neonatal infection risks after preterm premature rupture of membranes [61,62]. Antibiotics have not been shown to prolong pregnancy or improve neonatal outcomes in the case of preterm labor with intact membranes. Amoxicillin-clavulanic acid has even been associated with increased risk of neonatal necrotizing enterocolitis in these cases [61,62]. As with other antenatal interventions, specific data on antibiotic effectiveness in periviable births is scarce. Nevertheless, they are generally recommended for intrapartum group B streptococci prophylaxis and pregnancy prolongation after preterm premature rupture of membranes even at 22–24 weeks of gestation.

Magnesium sulphate has been shown to reduce the risk of death or cerebral palsy in preterm neonates born at 24 to 32 weeks [63,64]. It is generally recommended also in periviable births despite the lack of specific data for these gestational ages.

No randomized trials have been performed on fetal surveillance with cardiotocography and/or caesarean section in periviable births. There are observational data suggesting potential benefits of caesarean delivery of periviable neonates at more than 23 weeks of gestation in the case of fetal malpresentation [65,66,67,68]. These have, however, not been confirmed in all studies. At less than 22 weeks of gestation, caesarean section should only be performed for maternal indication (e.g., severe bleeding) [9].

Assuming that gestational age determination is reliable, current evidence suggests that neonatal survival is not possible before 22 completed weeks of gestation. After 22 weeks (22 0/7–22 6/7 weeks), the survival rate varies from 1 to 15%. According to the Slovenian National Perinatal Information System (NPIS), in the period of 2013–2021, only 1 out of 21 live-born neonates at this gestational age survived to hospital discharge at the tertiary perinatal center Ljubljana Maternity Hospital (survival rate 5.8%) [69]. Among survivors, the incidence of severe disability was high (87–98%). Survival (the short-term goal in these cases) cannot usually be achieved by any of the approaches, which include cesarean section, neonatal resuscitation, and intensive care. Given the greater risk of complications to the woman and the lack of benefit to the fetus and newborn, beneficence ethical principle does not require physicians to offer aggressive obstetric or neonatal care that includes cesarean section and/or neonatal resuscitation at 22–23 weeks of gestation [7,29,32].

At 23 weeks gestation (23 0/7–23 6/7 weeks; often called the “gray zone”), survival of neonates ranges from 8% to 56% [70,71]. Disease-free survival ranges from 7% to 23% worldwide. According to NPIS data for the period of 2013–2021, 46 infants were born alive at 23 weeks at Ljubljana Maternity Hospital, and 22 (48%) of them survived to discharge [69]. There is no convincing evidence that cesarean delivery improves fetal/neonatal outcomes at this gestational age, and it should not be routinely performed. If there is a fetal indication for a cesarean section, the surgery can only be performed if there is a plan to resuscitate the child after birth (although some call such an approach an “uncontrolled experiment”). Neonatal resuscitation can prevent impending death in a small percentage of newborns; however, death is still more common than survival at these gestational ages, and intact survival is rare. Immediate neonatal resuscitation is also not mandatory in these cases from the standpoint of justice, because, based on the evaluation of contemporary outcomes, a uniform aggressive obstetric and neonatal approach could lead to exploitation [7,29,32]. Nevertheless, we are faced with a growing number of maternity hospitals deciding to treat neonates born after the 23rd or even 22nd week of gestation [72,73,74,75]. Given the known negative impact of late initiation of intensive care, the decision not to withhold resuscitation and early intensive treatment seems reasonable. Some authors advise prenatal scoring which includes antenatal and immediate postnatal components as a decision tool for active resuscitation in the delivery room for infants born at 23 weeks gestational age. Such scoring includes also involves consulting parents before delivery and respecting their wishes [76]. If it turns out that the child has no chance of survival or that the outcome of treatment will be very poor, we can withdraw intensive treatment and start palliative (comfort) care at a later stage [77,78,79]. Such an approach can be justified by the fact that most extremely immature preterm infants die in the first days after birth. In contrast with intensively treated adult patients, however, each additional day in the intensive care unit improves the prospects for a neonatal survival [80].

At completed 24 weeks (≥24 0/7 weeks), unless the newborn has severe congenital anomalies or severe growth restriction, survival is the expected short-term outcome for most newborns, with up to 54% of infants surviving the neonatal period. According to NPIS data, 94 babies were born alive at the Ljubljana Maternity Hospital at 24 weeks between 2013 and 2021. Of these, 62 (66%) survived to discharge [69]. At the same time, there is a reasonable likelihood to achieve the long-term goal of low morbidity and high functional status, as the survival rate without severe sensory, motor, or cognitive impairment ranges from 12 to 38%. Less than 10% of children born at 24–25 weeks of pregnancy in Slovenia develop cerebral palsy [81]. These data imply the obligation to act (offer treatments) based on beneficence at completed 24 weeks of gestation [7,29]. Cesarean section based on fetal indication should be offered and even recommended. Neonatal resuscitation should be performed immediately after birth, and the newborn should be transferred to the neonatal intensive care unit. It is, however, necessary to obtain parental consent for all these interventions [29,31].

Approaches to life-sustaining interventions in periviable newborns (born at gestational age less than 25 + 0 weeks) is presented in Table 2.

Neonatal palliative care has its place in cases in which good outcomes of the treatment in the short and/or long term (i.e., survival without irreversible distressing disabilities) becomes very unlikely and continuation of treatment is no longer reasonable [77,82,83]. Such care aims to identify and meet the child’s basic needs. These include maintaining body temperature, relieving hunger/thirst and discomfort, and allowing contact with parents. For such newborns, the alternative of “supportive care”, in which life-sustaining measures potentially causing pain and suffering are withdrawn, is ethically justifiable [84,85,86]. Such decisions and the professional rationale for them must be clearly and compassionately presented to parents (who may demand that “everything should be done” to save their child) [87].

## 5. Discussion

Decisions on antenatal and intrapartum obstetric interventions as well as initiating, withholding, or withdrawing neonatal life support after periviable birth should only be taken after individualized parental counselling. This is optimally addressed by a multidisciplinary team that includes perinatologists, neonatologists, and potentially other experts, such as midwives, clinical psychologist, etc. Such a multidisciplinary approach also ensures the continuity of care after hand-offs as clinicians who have counselled the mother and her partner may not be the same ones providing obstetric and neonatal care during and after birth.

Parents should be given the opportunity to make decisions based on their values and preferences. However, such complex decisions can only be made after being presented with accurate and unbiased information on potential short-term and long-term neonatal and maternal outcomes. This is extremely challenging, as neonatal outcomes in periviable births cannot be predicted reliably due to several reasons. Numerous factors other than gestational age per se are associated with neonatal outcomes of periviable births [87]. These include fetal sex, plurality, and also birth weight which can only be estimated before birth [87]. It is important to acknowledge that additional information can become available after the initial conversation with the family. These information can change previous recommendations profoundly. Follow-up counselling should always be performed when new relevant data become available on maternal, fetal, or neonatal status.

Counselling on neonatal outcomes in periviable births is also challenging due to often biased information on which outcomes are assessed [3,4,31]. The outcomes reported in the literature come from different studies using different inclusion and exclusion criteria. Moreover, clinical practice and outcomes vary significantly among centers and countries [3,21,22,23,24,25,26,27,28]. As a result, reported outcomes cannot be simply generalized. In addition, neonatal outcomes change over time. A 2017 study showed a significant increase in both neonatal survival but also survival without neurodevelopmental impairment over time with advances in medical technology and clinical management [7]. Despite rates of neonatal death and long-term handicap remaining high, the increase in survival of neonates born at the limit of viability was not necessarily associated with more long-term neurologic injury [7].

Finally, it is important to consider the fact that most data on the effects of obstetric interventions on neonatal outcomes in periviable births are extrapolated from studies on preterm births at more than completed 26 weeks of gestation. The data on most interventions listed in Table 1 come from research conducted in 1970s and 1980s. Births at less than completed 24 weeks of gestation were generally not included in those studies. As a consequence, there is little specific evidence on the effectiveness of many antenatal interventions on neonatal outcomes at less than 24 weeks.

## 6. Conclusions

All clinicians have obligations to the fetus/neonate based on the principle of beneficence. We must balance these with our obligations to the parents, which are based on beneficence, trust, and respect of autonomy. Reliable prediction of outcome in the periviable period is not possible and the psychosocial aspects of each case are unique. Therefore, the decision-making process must be carefully coordinated and should involve perinatologists, neonatologists, and other medical specialists [88]. Optimal counselling of parents is extremely important. It includes a discussion between the parents and obstetricians as well as pediatricians/neonatologists who should provide up-to-date statistical data on the survival rates and risks of long-term sequelae. The consultation should also include some information about what to expect: both during birth, during neonatal resuscitation (if resuscitation is needed), during the dying phase (if it occurs), and about the child’s long-term illnesses (if they occur). Parents’ wishes are very important, and we must include them in all discussions on treatments before, during, and after birth [89,90,91,92].

Every pregnant woman, fetus, and newborn is different, and everyone should have the right to an individualized assessment before deciding on the extent of their treatment. In this regard, we must consider all principles stated in the paper as general recommendations that must be customized to individual cases. Despite the complexities of decision making, we must not forget that these decisions affect the entire life of the child and his or her family [78]. Many decisions (e.g., the decision to withhold or withdraw treatment) are also not reversible and may have long-term effects on the mental health of family members [93,94].

National guidelines and recommendations are based on arbitrarily set gestational age limits at which life-sustaining interventions and treatments are not recommended, can be considered, or are recommended. Such an approach neglects not only individual prognostic outcome factors (e.g., birth weight, gender, prenatal maturation, or neuroprotection treatments), but also center-specific factors (such as intensity of perinatal treatment and the attitude of healthcare professionals to periviable infants) [95,96].

## 7. Future Directions

Counselling in periviable births should respect the pregnant woman’s autonomy as well as ethical principles of beneficence, non-maleficence, and justice. In order to assure this, the risks of short-term and long-term neonatal complications following periviable births should be presented accurately and objectively. This cannot be achieved by simply extrapolating outcomes reported in the medical literature, as these are time-dependent and institution-dependent. In the future, parents faced with making complex decisions regarding antenatal, intrapartum, and neonatal management of periviable births should be presented with estimated risks calculated by taking into account factors such as exact gestational age, fetal/neonatal sex, plurality, birth weight, obstetric interventions (e.g., antibiotic prophylaxis, antenatal steroids, magnesium neuroprotection, mode of delivery, etc.). Importantly, these calculations should use contemporary institution-specific data. The development of electronic medical records and hospital information systems should allow outcome predictions that will allow such counselling and help parents make decisions based on their preferences and values.

## Figures and Tables

**Table 1 children-10-00784-t001:** Approaches to antenatal/perinatal care where birth is imminent or indicated at less than 25 weeks of gestation.

»Transport in utero«	due to considerably better prognoses (if neonate born at centers with equipment and expertise), it is recommended if preterm birth is likely and life-sustaining interventions are planned or may be a possibility not indicated if only palliative care is planned always aim for transfer of pregnant woman unless transfer puts mother’s life at risk consult with higher level service as required
Tocolysis	to allow administration of steroids to achieve »transport in utero« consider individual benefits vs. risks of delaying birth (e.g., maternal infection, placental abruption)
Steroids	associated with decreased neonatal mortality/morbidity recommended from 22 + 0 weeks if high risk of preterm birth and life-sustaining interventions are planned administered prior to »transport in utero« if possible, 48 h prior to birth
Antibiotics	shown to prolong pregnancy and reduce neonatal infections after preterm premature rupture of membranes indicated for group B streptococci prophylaxis recommended from 22 + 0 weeks if high risk of preterm birth and life-sustaining interventions are planned
Cardiotocography (CTG)	not recommended before 24 + 0 weeks little evidence for interpretation of CTG and thus limited usefulness between 24 + 0 and 28 + 0 weeks fetal physiology/individual circumstances/clinician expertise should be taken into account when interpreting CTG
MgSO_4_	protects gross motor function, reduces the risk of cerebral palsy if given shortly before birth recommended if birth is imminent and life-sustaining interventions are planned or may be a possibility
Cesarean section (C.S.) for fetal indication	at extremely low gestational ages, advantages of C.S. for the infant are inconclusive and conflicting in specific circumstances C.S. may be considered (multiples, breech presentation, obstetric history, future pregnancy, parental wishes) consensus: for fetal indication not recommended before 24 + 0 weeks

**Table 2 children-10-00784-t002:** Approaches to life-sustaining interventions in periviable newborns born at a gestational age less than 25 weeks.

Gestational Age (GA) (Weeks)	Recommendations
**Unknown/Uncertain**	initiate life-sustaining interventions when GA/clinical situation becomes clearer, decision depending on the situation is made
**<23 + 0**	aggressive obstetric care/cesarean section is highly discouraged life-sustaining interventions are not recommended palliative care is acceptable and is advocated
**23 + 0–23 + 6** (»gray zone«)	cesarean section is not routinely performed; however, it is acceptable for fetal indication immediate resuscitation of the newborn is not mandatory; however, it is acceptable and reasonable informed parental consent is obligatory
**24 + 0–24 + 6** (without malformation, severe SGA)	cesarean section for recommended fetal indication immediate resuscitation of the newborn/admission to NICU is encouraged parental consent is desirable

## Data Availability

No new data was collected for the purpose of this review.

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
