# Peer review of "Ethical Dilemmas in Neonatal Care at the Limit of Viability"

_children, 2023, doi:10.3390/children10050784_

Round 1
Reviewer 1 Report
This is a nice review with the survival and morbidities data from own population. All given data from the references are appropriately discussed. The paper represents local attitudes combined with general approach to the patient at the border of viability and his family.
Author Response
Dear reviewer, we are very grateful for your encouraging evaluation of our paper.
Reviewer 2 Report
This is well written and paper appropriately presenting the topic. The statements are clear, adequate, and helpful for everyday clinical practice.
Author Response
Dear reviewer, we are very grateful for your valuable review. According to your comment we have re-checked English language and corrected several errors in the text.
Reviewer 3 Report
Thank you for requesting to provide a review of this article, which has a subject of high interest.
The main purpose of the analysis was to evaluate the dimension of periviable birth management.
The main question adressed in the research was whether there is a beneficence-based obligation to offer aggresive obstetric or neonatal management in fetuses born at lower gestational ages than 24 weeks.
The study is a retrospective review. The topic is original and relevant in the field and brings usefull knowledge regarding the subject. A comprehensive search strategy was used and so. The review methodology was comprehensive with screening and data extraction. When it comes to the methodology used, no specific improvements should be considered from my point of view.
The conclusions are consistent with the evidence and the arguments presented, and they adress properly to the main question which conducted the analysis.
The references are appropriate and well suited for this kind of study.
Regarding the structure and accuracy of the phrases, the manuscript has well structured information, with supported evidence and well structured phrases.
The manuscript is original and well defined. The results provide an advance in current knowledge. The results are being interpreted appropriately and are significant, as well as the conclusions.
The article is written in an appropriate way.
The study is correctly designed and the analysis is being performed at high standards, so the data are robust enough to draw the conclusion.
Surely the paper will attract a wide readership.
The English language is appropriate and well understandable.
There are only 3 things to add in the lines below, but the article should be published after the corrections are made:
Line 30: towards, not „toward”
Line 33: compared to, not „compared with”
Line 48: in the management, not „in management”
Author Response
Dear reviewer, we are very grateful for your very detailed assessment of the paper and all the comments. We have corrected the mistakes and English language accordingly to your comments.
With respect to Academic Editor Comments we have added 2 tables, expanded the content with the new information, new references have been also added.
Reviewer 4 Report
This scientific research work deals with a very interesting topic of importance for gynecologists, obstetricians, neonatologists, pediatricians, and last but not least, parents, as a unique team in making the best decisions about the approach and treatment of premature infants, in order to achieve the best possible outcome. The paper also addresses and explains the ethical concepts and dilemmas that the aforementioned doctors, especially those in tertiary and lower-level health institutions, often encounter in their clinical work. Respect and implemention of described ethical concept when approaches pregnant women and premature newborn is important and necessary in the modern approach and antenatal screening of pregnant women.
The paper also presents data on the survival rate of premature infants depending on gestational age, as well as clinical approach guidelines based on gestational age, which is of great importance when taking an empirical approach in countries where similar studies have not been conducted.
This work is significant for medical and social fields, and I believe it should be published in its entirety.
Author Response
Dear reviewer, we are very grateful for your review of our paper. According to your comments we have re-checked English language and made several corrections.
Reviewer 5 Report
This thoughtful and informative paper by Cerar et al makes essential contributions to our understanding of the very important topic of ethical dilemmas in caring for extremely premature infants at the limit of peri viability. The paper adequately mentions the guidelines from various international organizations and the ethical dilemmas faced by medical professionals and families of such patients. The paper could be strengthened by the minor suggestions listed below.
Page 3- line 111-113: Please cite the references for the data mentioned.
Page 4- line 122-123: Please cite the references for the data mentioned.
Author Response
Dear reviewer, we are very grateful for your review and the comments. We have added the missing reference, re-checked English language and made several corrections.
According to Academic Editor Comments we have also added 2 tables, expanded the content with the new information, new references to related paper have been also added.